# Injuries during Laparoscopic Cholecystectomy: A Scoping Review of the Claims and Civil Action Judgements

**DOI:** 10.3390/jcm10225238

**Published:** 2021-11-10

**Authors:** Roberto Cirocchi, Laura Panata, Ewen A. Griffiths, Giovanni D. Tebala, Massimo Lancia, Piergiorgio Fedeli, Augusto Lauro, Gabriele Anania, Stefano Avenia, Salomone Di Saverio, Gloria Burini, Angelo De Sol, Anna Maria Verdelli

**Affiliations:** 1Department of Medicine and Surgery, University of Perugia, 06132 Perugia, Italy; roberto.cirocchi@unipg.it (R.C.); massimo.lancia@unipg.it (M.L.); stefanoavenia1@gmail.com (S.A.); 2Legal Medicine and Insurance Office, Santa Maria della Misericordia Hospital, 06129 Perugia, Italy; laurapanata@gmail.com (L.P.); annamaria.verdelli@gmail.com (A.M.V.); 3Department of Upper Gastrointestinal Surgery, Queen Elizabeth Hospital Birmingham, University Hospitals Birmingham NHS Trust, Birmingham B15 2GW, UK; ewen.griffiths@uhb.nhs.uk; 4Institute of Cancer and Genomic Sciences, College of Medical and Dental Sciences, University of Birmingham, Birmingham B15 2TT, UK; 5Surgical Emergency Unit, John Radcliffe Hospital, Oxford University NHS Foundation Trust, Oxford OX3 9DU, UK; giovanni.tebala@ouh.nhs.uk; 6School of Law, Legal Medicine, University of Camerino, 62032 Camerino, Italy; piergiorgio.fedeli@unicam.it; 7Department of Surgical Sciences, Hospital “Policlinico Umberto I”, “Sapienza” University of Rome, 00161 Rome, Italy; augusto.lauro@uniroma1.it; 8Department of Medical Science, University of Ferrara, 44121 Ferrara, Italy; ang@unife.it; 9Department of General Surgery, ASUR Marche, AV5, Hospital of San Benedetto del Tronto, 63074 San Benedetto del Tronto, Italy; salo75@inwind.it; 10Department of General and Emergency Surgery, Hospital “Ospedali Riuniti di Ancona”, 60126 Ancona, Italy; 11Department of General Surgery, St. Maria Hospital, 05100 Terni, Italy; a.desol@aospterni.it

**Keywords:** biliary injury, laparoscopic cholecystectomy, postoperative complications, litigation, legal practice

## Abstract

Background. To define what type of injuries are more frequently related to medicolegal claims and civil action judgments. Methods. We performed a scoping review on 14 studies and 2406 patients, analyzing medicolegal claims related to laparoscopic cholecystectomy injuries. We have focalized on three phases associated with claims: phase of care, location of injuries, type of injuries. Results. The most common phase of care associated with litigation was the improper intraoperative surgical performance (47.6% ± 28.3%), related to a “poor” visualization, and the improper post-operative management (29.3% ± 31.6%). The highest rate of defense verdicts was reported for the improper post-operative management of the injury (69.3% ± 23%). A lower rate was reported in the incorrect presurgical assessment (39.7% ± 24.4%) and in the improper intraoperative surgical performance (21.39% ± 21.09%). A defense verdict was more common in cystic duct injuries (100%), lower in hepatic bile duct (42.9%) and common bile duct (10%) injuries. Conclusions. During laparoscopic cholecystectomy, the most common cause of claims, associated with lower rate of defense verdict, was the improper intraoperative surgical performance. The decision to take legal action was determined often for poor communication after the original incident.

## 1. Introduction

Since its introduction, laparoscopic cholecystectomy has rapidly become the gold standard for the treatment of gallstones, due to the significant improvement in overall morbidity, shorter hospital stay, and prompt return to work. At the moment, laparoscopic cholecystectomy (LC) is one of the most frequently performed procedures worldwide in general surgery [1], but biliary duct injuries (BDI) still represent the most significant complication following laparoscopic cholecystectomy [2]. Studies have reported an increase in BDI up to 0.8% with minimal access surgery, which is much higher than the previously reported rates for open cholecystectomy, which was between 0.2–0.3% [3]. Risk of BDI in pediatric patients is even higher (0.5–2.5%), due to the increasing incidence of gallstones and subsequent laparoscopic cholecystectomy in children, and possibly also due to the suboptimal experience of pediatric surgeons with this technique [4].

However, this procedure has shown a consistent increase of the number of insurance claims. This implicates high costs for the health service [5].

During laparoscopic cholecystectomy, one of the common causes of biliary injuries is the lack of clear identification of the structures inside the hepatocystic triangle, where it is possible to misidentify the common bile ducts or aberrant right duct as cystic ducts, and wrongly divide them. This can also occur to the right hepatic artery, which can be misidentified as cystic artery and divided [6].

For these reasons, it is crucial to identify the safe zone of dissection to avoid inadvertent vasculobiliary injury. The identification of the safe zone (B-SAFE) is based on the following fixed anatomical landmarks: common hepatic/bile duct and base of hepatic segment 4 (B), Rouviere’s sulcus and segment 4 (S), hepatic artery (A), umbilical fissure (F), and enteric viscera (E) [7].

This zone lies above a R4U line running from the upper edge of the Rouviere’s sulcus (R) to the umbilical fissure (U) across segment 4 [4]. This R4U line divides the safe zone, for dissection, located above the line from the dangerous one located below the line [8].

The key anatomical landmark is the Rouviere’s sulcus, which is located on the surface of the right lobe of the liver, running to the right of the hepatic hilum, and of variable length. In around 80% of patients, it is clearly visible as a fully open or partially open congenital defect in the liver. The sulcus contains the structures of the right portal pedicle. The entire dissection must be performed above this imaginary line joining the sulcus and the base of segment 4 of liver in order to avoid injury [8].

In the open surgery era, BDI were categorized using the Bismuth classification (1982), based on the location of the injury in the biliary tract. BDI were categorized according to the distance from the hilar structure, especially the bile duct bifurcation, the involvement of bile duct bifurcation, and the individual right sectoral duct [9].

This Bismuth classification does not encompass the whole spectrum of injuries that are possible with open and laparoscopic cholecystectomy, mostly because it was originally meant only for benign biliary strictures. On the contrary, in laparoscopic cholecystectomy era, BDI are more heterogeneous and severe than those in open cholecystectomy.

Strasberg and colleagues have described a more relevant classification, similar to the one from Bismuth, but which allows the differentiation between small (bile leakage from the cystic duct or aberrant right sectoral branch) and more serious injuries of the biliary tree occurring during laparoscopic cholecystectomy [10].

The limitation of Strasberg’s classification is that it does not include additional vascular injuries, which can increase the severity of the injury, can be associated with focal ischemia of the liver and biliary tree, and complicate the surgical management of BDI [11].

Other BDI classification systems have been developed, including those of Bergman, Neuhaus, Csendes, and Stewart, which include the other additional possible injury types. [12,13,14,15].

Furthermore, the European Association for Endoscopic Surgery held a consensus conference on BDI with the purpose to produce a definitive classification system. In this system, BDI are classified into three categories: anatomic, time of injury, and mechanism (ATOM) [16].

Delayed recognition of BDI may play a role in the increased incidence of litigation following LC. In fact, the majority of BDI often are not diagnosed at the time of operation. As a consequence, the medicolegal issue is associated with a more expensive litigation. Furthermore, from the clinical point of view, an early treatment of BDI leads to a better outcome [17].

Assessment and evaluation of the risk of BDI represents an active risk management to help improving the standards of care and decreasing intraoperative mishaps, as well as the chances for claims. Our hypothesis is that the knowledge of the most frequent causes for legal claims related to BDI may help the surgeon to focus his or her attention on the critical steps of the pathway of patients with gallstones.

The right to safety of care is an integral part of the broader right to health. In Italy, Law 24/2017 has modified professional responsibility of the healthcare sector [18]. The dispute for alleged malpractice is one of the specific issues of clinical risk management, because it is a potential source of economic damage for the healthcare facility. Clinical risk management strategies are one of the tools that all healthcare professionals should adopt [19]. Through this system, healthcare facilities are responsible for the continuous improvement of healthcare standards to facilitate the achievement of clinical excellence. Integration between monitoring, clinical audits, adverse event reporting, and litigation management activities are strongly recommended.

We performed a review of medicolegal claims related to LC-related injuries during surgery for gallbladder stones in the international literature over the past three decades, in order to identify the most crucial surgical steps of the treatment associated with the highest incidence of very expensive medico-legal claims, and to try to find surgical and medical solutions to prevent or limit them.

## 2. Materials & Methods

In this scoping review, the O’Malley and Arksey six-stage methodological framework was performed: identifying the research question, searching for relevant studies, selecting studies, charting the data, collating, summarizing, and reporting the results, and consulting with stakeholders to inform or validate study findings [20].

The research question of this scoping review was “which are the injuries during laparoscopic cholecystectomy associated with medicolegal claims and civil action?”. The following search strategy was carried out using the following keywords:Claim OR malpractice litigation OR legal practice.Laparoscopic cholecystectomy OR Biliary Duct InjuryA combination of points 1 AND 2.

The search for relevant studies was performed on PubMed, SCOPUS, Web of Science, and Google Scholar, for studies dated from January 1991 to January 2021. The checklist of PRISMA-ScR (Preferred Reporting Items for Systematic reviews and Meta-Analyses extension for Scoping Reviews) was performed [21].

No language restriction was applied. When the articles were published by the same study group and an overlap was found, only the most recent article was included in order to avoid duplication of data. The PubMed function “related articles” was used to extend the search. We also performed hand-search of references of included studies, to identify other potentially eligible sources. In this analysis of grey literature, another search was performed on Google.

We included all articles that reported medico-legal claims for laparoscopic cholecystectomy. All studies were independently assessed for eligibility by two reviewers (P.F. and R.C). Any controversy was resolved by a consensus among all the reviewers, if possible.

The following data were independently extracted by two authors (R.C and B.T.): first author name, study design, country, sample size, objective, inclusion and exclusion criteria, and outcome measures.

The identified studies were summarized according to key themes, based on similarities of their main intervention and metrics. Compared with the methodology of a systematic review, a scoping review of the risk of bias evaluation of included studies was not performed.

The thematic framework was the following:Step of the care pathway associated with claims and civil action;Location of injuries associated with claims and civil action;Type of injuries associated with claims and civil action.

In the analysis of the steps of care associated with claims and civil action, there was an important limitation for the heterogeneous different type of outcomes and few definitions reported in included articles. For this reason, we have aggregated similar conditions reported in the literature in homogeneous groups.

The quality assessment of studies was performed through the Risk Of Bias In Non-randomized Studies of Interventions (ROBINS-I) assessment tool [22,23].

## 3. Results of Scoping Review

The PRISMA flow chart for systematic review is presented in Figure 1. The initial search produced 646 potentially relevant articles. After the titles and abstracts were screened for relevance, 22 remaining articles were further assessed for eligibility, and 8 were excluded. The 14 studies, whose characteristics are reported in Table 1, are included in this scoping review [17,24,25,26,27,28,29,30,31,32,33,34,35]; the reasons of exclusion of the 8 studies are reported in Figure 1 [36,37,38,39,40,41,42,43].

In the 14 studies, 2406 patients were enrolled. The higher number of studies was performed in the U.K. (6 studies, 1104 patients) and the U.S.A (6 studies, 1148 patients), with only two studies being performed in other nations (14.2%): The Netherlands (133 patients) and Turkey (21 patients). The majority of the studies performed in the U.K. reported data from the National Health Service Litigation Authority (NHSLA) for England (5 studies, 953 patients); on the contrary, the databases utilized in American studies were very heterogeneous. The range of publications was very wide between 1998 and 2020, and only half of the papers were published in the last ten years.

## 4. Quality Assessment of Studies

A total of 15 Non-Randomized Studies (Non-RCT) were identified and analyzed with the Risk Of Bias In Non-randomized Studies of Interventions (ROBINS-I) assessment tool [22,23] (Figure 2 and Figure 3).

Nine out of fourteen studies were assessed as “moderate” risk of overall bias [5,17,24,25,26,28,29,32,35], while two were determined to have a “serious” risk [27,31], and three a “critical” risk [30,33,34]. Regarding the bias due to confounding, nine out of fourteen studies were evaluated as having a “moderate” risk [5,17,24,25,26,28,29,32,35]; differently this risk was higher as “serious” in studies performed through insurance surveys or databases [31,34], or “critical” in single team experiences [30,34] or Justice Court [27]. In our opinion, large national databases reduce the risk of bias due to confounding. Furthermore, bias in the selection of participants was similar to the previous domain; the difference was the higher risk of Perera et al., (single hospital experience) and the lower risk of De Reuver et al. (insurance database). For domains 3, 4, 5, and 7, the judgement was particularly homogeneous. However, domain 6 was particularly heterogeneous for the no information reported or different methods used to assess outcomes in different intervention groups.

## 5. Step of the Care Pathway Associated with Claims and Civil Action

### 5.1. Causes of Litigations during Laparoscopic Cholecystectomy

This analysis of the dataset was performed through the radar chart (Figure 4) that compared the multiple quantitative variables (Figure 5). The most common step of care associated with litigation after laparoscopic cholecystectomy was the improper intraoperative surgical performance (Table 2): 47.6% ± 28.3% (mean ± Standard Deviation), followed by improper post-operative management of the injury: 29.3% ± 31.6%. Very uncommon issues were incorrect presurgical assessment (6.2 ± 2.3) and unnecessary surgery (3.6% ± 2.1%).

The improper intraoperative surgical performances were commonly consequent to “problematic visualization” (42.83% ± 34.34%); other uncommon conditions of improper intraoperative surgical performances were the following: improper response to damage (25.25% ± 17.8%), inadvertent visceral damage (17% ± 18.6%), and failure to convert to open surgery (14% ± 13.5%) (Table 3, Figure 6).

### 5.2. Verdicts in Litigations during Laparoscopic Cholecystectomy

Data about the verdicts (plaintiff victory or defense verdicts, arbitration, settlement) were reported by a small number of studies, and the analysis was performed with the radar chart (Figure 7).

The highest rate of defense verdicts (Figure 8) was reported in claims for improper post-operative management of the injury (69.3% ± 23%). A lower rate of defense-positive verdicts was reported in the incorrect presurgical assessment (39.7% ± 24.4%), and in the improper intraoperative surgical performance (21.39% ± 21.09%).

The “problematic visualization” is the most frequent allegation of negligence with the lower rate of defense verdicts (17.4 ± 21.2) (Figure 9); however, the inadvertent visceral damage during cholecystectomy was associated with the higher rate of defense verdicts (47.75 ± 36.48) (Figure 10). Few data about verdicts were reported about the “failure to convert to open surgery” and “improper response to damage”.

### 5.3. Location of Injuries in Litigations Performed for Laparoscopic Cholecystectomy

This outcome was reported in eight studies (1185 claims), and the injuries were located at the bile duct, vascular system, and bowel (Figure 11). The highest number of injuries were located at the biliary duct (89.8%-mean), but 30.6% (mean) of cases reported a generic biliary duct injury without reporting a precise description of the location (Figure 12). In biliary duct injuries, the most frequent location was the common bile duct (25.5% ± 29%); other lesions were located in the hepatic duct (15.2%) and the cystic duct (7.5% ± 7.1%) (Figure 13).

A defense verdict was more common in cystic duct injuries (100%) [25], but was less likely in hepatic duct (42.9%) [25] and common bile duct injuries (10%) [25] (Figure 14).

Other uncommon locations injured were bowel and vessels. The incidence of bowel injuries was 7 ± 4%. The rate of plaintiff verdicts and defense verdicts was the same (50%).

Vascular injuries were reported (6 ± 3.2%). In all claims and civil action for VLC, vascular injuries were reported at the level of iliac arteries (10.9%) [25], portal vein (2.2%) [25], and lacerated hepatic artery (6.2%) [25]. In vascular injuries of the iliac artery, the rate of defense verdicts was higher (60%) [25].

## 6. Type of Injuries Associated with Claims and Civil Action Judgments

The most common injuries were the cut/transection of biliary duct (29.63–56.75%) [30,33], and the resection of Biliary duct (40.5–61.11%) [31,33]. Uncommon injuries were the clipping of biliary duct (9.26%) [30] and burn of biliary duct (8.1%) [33].

## 7. Discussion

The number of laparoscopic cholecystectomies performed every year is particularly high, thus the incidence of associated injuries is constantly significant; for this reason, claims for cholecystectomy are still one the most common causes of legal litigation, and represent a serious problem for surgeons.

This scoping review is the first systematic analysis of the claims consequent to laparoscopic cholecystectomy as published in the medical literature.

In our analysis, the critical phase of care during laparoscopic cholecystectomy was an improper intraoperative surgical performance, which is the most common cause of claims (mean ± Standard Deviation: 47.6 ± 28.3) (Figure 3), and is associated with the lowest rate of defense verdicts (mean ± Standard Deviation: 21.39 ± 21.09) (Figure 4). In effect, the highest incidence of error is during the intraoperative phase, and these errors can be associated to negligent surgical behaviors.

The most common cause of improper intraoperative surgical performances was a “problematic visualization” (42.83 ± 34.34), which was schematized by Gordon-Weeks et al. as follows [39]:“Not performing a cholangiogram”;“Surgeons misconception about biliary anatomy”;“Poor dissection—no critical view obtained”.

Poor visualization and undefined anatomical landmarks during cholecystectomy can lead to disastrous consequences [44], whereas the role of intraoperative cholangiogram is still a matter for debate. For this reason, surgical societies have published guidelines and held consensus conferences, with the aim of providing a framework for the prevention of bile duct injuries during laparoscopic cholecystectomy. These studies were primarily focused on determining the optimal strategy to avoid bile duct injuries during the different steps of LC. In turn, this point can be categorized as suggested from Strasberg [45] and van de Graaf [46] in the Critical View of Safety Laparoscopic Cholecystectomy (CVS), that it is the best method for identification of the artery and cystic duct before their clipping and division. It is not a technique for dissection of these structures, but a way of avoiding misidentification and a catastrophic BDI [10,47,48].

The CVS is based on three fundamental principles:clearing of the Calot’s triangle by removing fat and fibrous tissue;exposing the lower third of the liver bed by separating the gallbladder from the cystic plate;after this dissection, only two tubular structures (cystic duct and cystic artery) should remain and can be seen entering the gallbladder [49].

To fulfil all three criteria, it is necessary to exert a proper gallbladder retraction for the adequate exposure of the anterior and the posterior aspects of the triangle of Calot [50].

Beyond the CVS, there are several other accepted techniques used to reduce the risk of BDI, such as the infundibular view technique and the fundus-first technique; however, both can be misleading, and represent error traps for the surgeon [51].

In the infundibular technique the putative cystic duct is dissected circumferentially, but unfortunately the same view can be obtained when the common bile duct/common hepatic duct (CBD/CHD) adheres to the edge of the gallbladder, while the real cystic duct is hidden. The dissection is performed around the CBD/CHD rather than the cystic duct (error trap), and this may lead to the typical biliary duct injury, particularly in a “hostile gallbladder” during cholecystectomy [49,50,52].

In the fundus-first technique, the dissection of the gallbladder from the liver bed is performed top-down, and the division of the cystic artery and cystic duct is postponed to the end of the dissection [53,54]. Unfortunately, severe inflammation of the cystic plate can lead the surgeon towards the wrong plane, with the risk of injuring the right portal pedicle and other hilar structures [55].

Intraoperative factors predictive for difficult CVS are dense peri-gallbladder adhesion, fibrotic scarring in the hepato-cystic triangle and gallbladder bed with a small shrunken gallbladder (so called scleroatrophic gallbladder), oozing during the dissection, multiple gallbladder perforations, and Mirizzi Syndrome. Several scores have been developed to predict a difficult cholecystectomy and the likelihood of conversion to open surgery (CLOC Score) [56].

In some complex cases, the anatomic identification of the cystic structures is just not possible, and the risk of biliary injury is very high. It is of the utmost importance to understand when the local conditions represent an unacceptable risk. In this scenario, the surgeon has five potential bail-out options [57,58,59]:abandon the procedure;convert to open procedure;cholecystostomy tube placement before abandoning the cholecystectomy;subtotal cholecystectomy (either by closing the remnant of the gallbladder neck or leaving it fenestrated;fundus first cholecystectomy.

The best choice depends on the specific clinical situation and, mostly, on the surgeon’s skill.

Performing a subtotal cholecystectomy involves the removal of as much gallbladder as safely as possible, with the lower portion closed (reconstituted) or left open (fenestrated). This entails a high risk of postoperative bile leakage from the cystic duct, in particular in case of spasm of the Oddi’s sphincter or stones in the CBD. In both cases, it is of vital importance to clear any stones from Hartmann’s pouch to avoid recurrent symptoms.

The concept of time-out (stopping rules) is always important for the surgeon, especially when facing a difficult situation. In this case, options include stopping the procedure, takings a break to obtain a clear judgment of the situation and reorientate the anatomical landmarks (B-Safe), or asking for help from a more experienced or HPB colleague [52].

In difficult and hostile gallbladders, the surgeon must be aware of the potential risks and dangers, in particular if the surgeon is not able to bail out in time and keeps going, instead of adapting the procedure to the safety of the patient [60].

The main point is that the operation must be stopped before the point of no-return, that is, after a severe complication has occurred or after the division of the cystic duct or artery. Therefore, the surgeon must be able to recognize the zone of the greater risk by identifying the following operative clues: severe adhesion, severe acute inflammation, a large impacted stone in the neck of gallbladder, and Mirizzi Syndrome [61].

Which strategies can the operator adopt in a critical situation? 

Calling for help: misidentification by false mental image [44,62] is a major cause of injuries of vascular and biliary structures. A more expert surgeon not involved in the operation since the beginning can be of great help in identifying the anatomical landmarks and prevent complications.

Use of intraoperative imaging:Intraoperative cholangiogram (IOC) is a safe technique that can be easily performed during laparoscopy, and is able to detect anatomical abnormalities and asymptomatic biliary stones, but needs specific skills, time, and costly equipment [63];Laparoscopic ultrasound is quick, safe, and non-invasive, but less accurate than IOC, and has a demanding learning curve [64];Near infrared fluorescent cholangiogram is a new, cheap, quick, and safe technique, but still under evaluation [65,66].

A time-out must always be taken after entering the abdomen, before dissecting the HC triangle, in case an anomalous anatomy is encountered, and before clipping and dividing the cystic artery and duct when the CVS has already achieved [7].

The common aim of these recommendations is the universal adoption of a “culture of safety in cholecystectomy” (COSIC). The landmarks of the COSIC, as recently reported from Vishal Gupta [61], are the following: (1) a clear understanding of relevant anatomy; (2) appropriate and timely use of bailout techniques; (3) obtaining CVS prior to division of cystic duct and artery in every case; (4) recognizing the importance of time-out; (5) use of intraoperative imaging; (6) obtaining a second opinion in difficult cases; and (7) importance of proper documentation.

Although some societies (Society of American Gastrointestinal and Endoscopic Surgeons—SAGES) performed campaigns to raise awareness among surgeons to use this philosophy in the “Safe Cholecystectomy Program” in order to minimize the risk of BDI (https://www.sages.org/safe-cholecystectomy-program/, accessed on 5 June 2021), there is still a low adoption of these suggestions [46], and the surgeons that non-use of CVS have a higher incidence of BDI (54.6%) than use CVS (25.8%) [67].

The reasons of this lacunar implementation of these recommendations are not yet clear.

Furthermore, the analysis of causes and prevention of laparoscopic BDI reported that errors leading to BDI are usually consequent to misperception, and not to errors in skill, knowledge, or judgment [44]. In many cases, surgeons do not recognize the intraoperative problem due to misperception, and not due to negligence [68]. These misperception errors are most common during the intraoperative phase, when the stress of the surgeon is high and physiological “signals” (anatomy) are obscured by “noise” (fat, blood, inflammation) [69]. For these reasons, it is very important to accelerate the development of training programs in nontechnical skills [70] related to surgical outcomes and patient safety.

This hypothesis was confirmed by some studies in which senior surgeons were not able to adequately reproduce the steps of the CVS. In actual fact, a study performed on 1108 consecutive patients undergoing laparoscopic cholecystectomy showed a disagreement between the opinions of the operators (80% of the surgeons in the analysis stated to carry out the CVS) and the video analysis of the interventions (10.8%) [71].

Iatrogenic injuries of the biliary duct entail a very high chance of legal action for malpractice, due to the very serious consequences associated to this unpleasant complication, such as bile peritonitis, sepsis, multiple organ injury, cholangitis, and liver abscess which, if not managed properly, can lead to death [41].

These incidents have profound effects on the lives of the patients and their families with long-term effects on work, social life, and family relationships [72]. Furthermore, these complications also have a highly detrimental effect on the surgeon who is defined as the “second victim” [73].

It is therefore important to improve the surgical techniques to reduce complications and litigations after laparoscopic cholecystectomy. However, only a very small proportion of such injuries arise from errors regarding the technical skills; unfortunately, BID is bound to remain a significant risk of this procedure from which nobody is immune, even experienced surgeons [41].

In fact, the decision to take legal action is determined not only by the original injury, but often for unempathetic relationship and poor communication after the original incident [72]. As strongly suggested by the UK General Medical Council, all professionals have a duty of candor towards the patients in particular when things go wrong [74].

The most important limitation of this study is a bias due to the high number of studies performed in Anglo-Saxon countries (85.8%) whose legal system is based on Common Law. This system is in use in the whole United Kingdom, and forms the basis of jurisprudence in the United States of America (with the exception of Louisiana), and in many other Commonwealth countries. This high prevalence of studies from Common Law countries, with respect to countries whose legal system is based on the Codex Iustinianeum, might have created a selection bias, and skewed the impression of the reader to a more lenient attitude of the judges towards these complications. Moreover, estimating the number of medical claims for BDI during laparoscopic cholecystectomy only on the basis of the analysis of the medical literature may provide a limited picture. A wider study taking into account the statistics of each legal system and the archives of worldwide civil and criminal courts might be much more reliable, but hardly feasible.

## 8. Conclusions

Our analysis highlighted that the most critical step of care during laparoscopic cholecystectomy was an improper intraoperative surgical performance, that is, the most common cause of claims which is associated with the lower rate of defense verdicts. The highest incidence of errors happen during the intraoperative phase, and these errors can be associated with negligence. The most common cause of improper intraoperative surgical performances was a “problematic visualization”.

Errors leading to bile duct injuries are often consequent to misperception, and not to errors of skill, knowledge, or judgment. In many cases, surgeons do not recognize the intraoperative problem due to misperception, and not due to negligence. These misperception errors are most common during the intraoperative phase, in which stress is at its highest.

Strategies to minimize claims would improve patient, care and reduce the litigation burden.

## Figures and Tables

**Figure 1 jcm-10-05238-f001:**
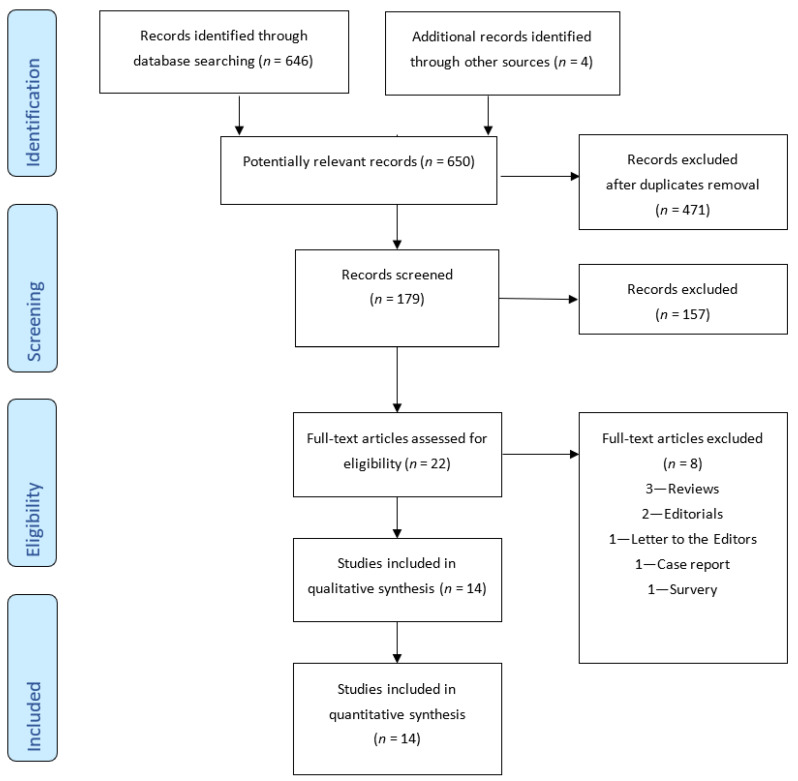
PRISMA flow diagram of study search.

**Figure 2 jcm-10-05238-f002:**
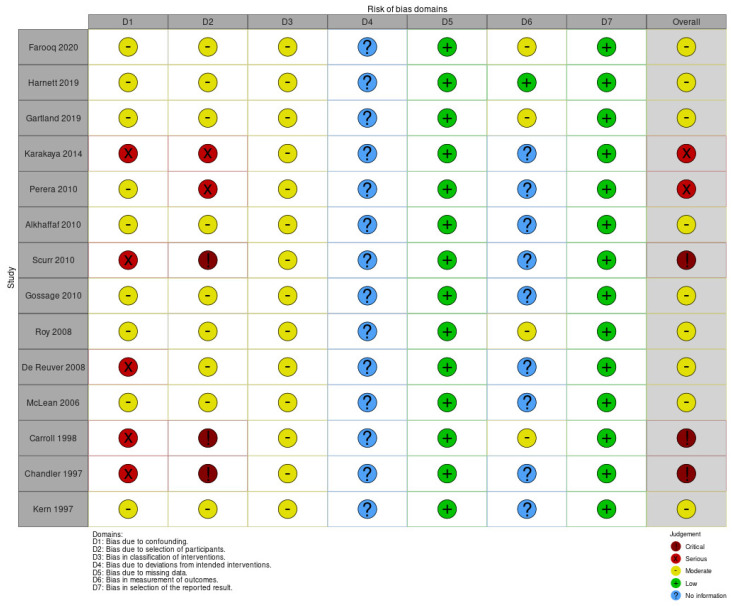
ROBINS-I risk of bias assessment summary: authors’ judgements about each methodological quality item for each included study.

**Figure 3 jcm-10-05238-f003:**
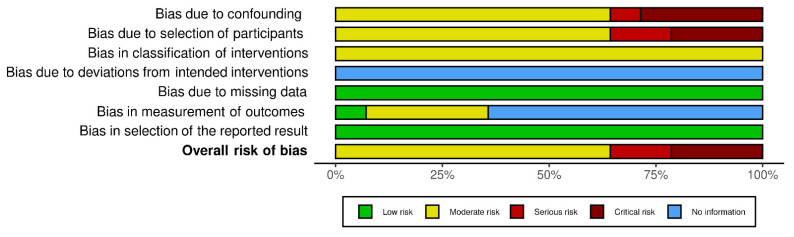
ROBINS-I risk of bias assessment graph.

**Figure 4 jcm-10-05238-f004:**
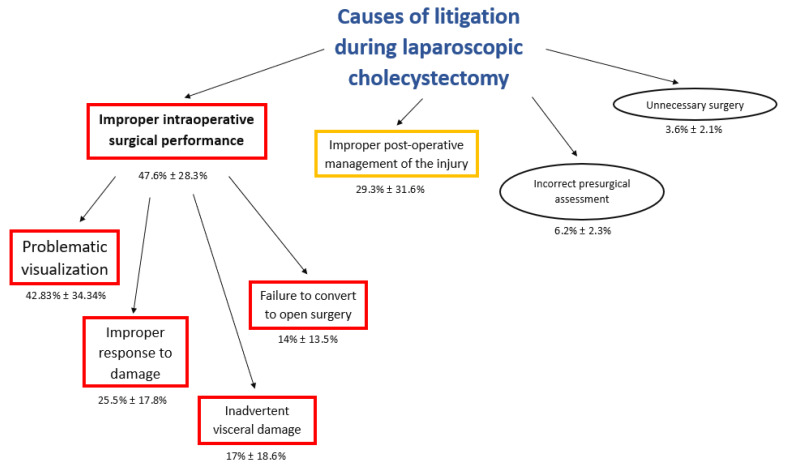
Flowchart of causes of litigation during laparoscopic cholecystectomy.

**Figure 5 jcm-10-05238-f005:**
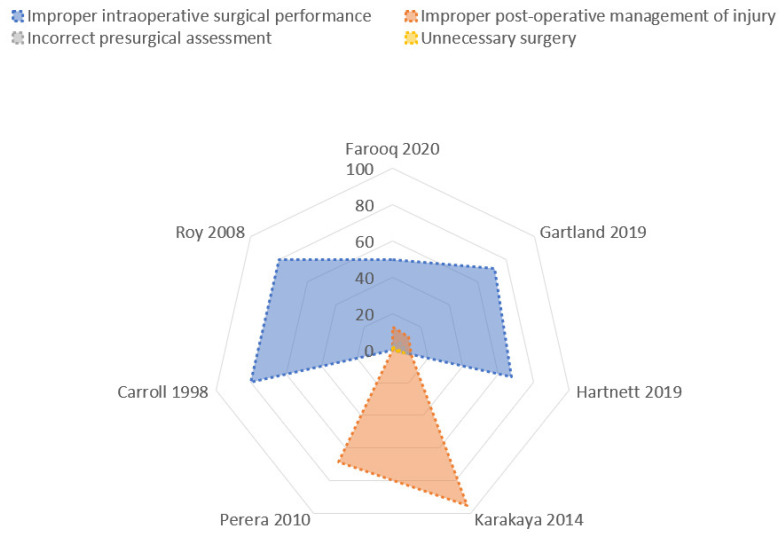
Radar chart: causes of litigations. The Radar chart shows two polygons, demonstrating the most common reasons of litigations (blue polygon: improper intraoperative surgical performance; pink: improper post-operative management injury). The blue polygon has the largest area, showing that, in effect, the improper intraoperative surgical performance is the most common cause of litigations. The smaller pink polygon is very asymmetrical for the heterogeneity of the rate, as reported in Table 3. The other two causes (incorrect presurgical assessment or unnecessary surgery) are very uncommonly reported, and the rate of occurrence is particularly low in the studies that analyzed this data; for these reasons, these two polygons (green and yellow) are not presented here.

**Figure 6 jcm-10-05238-f006:**
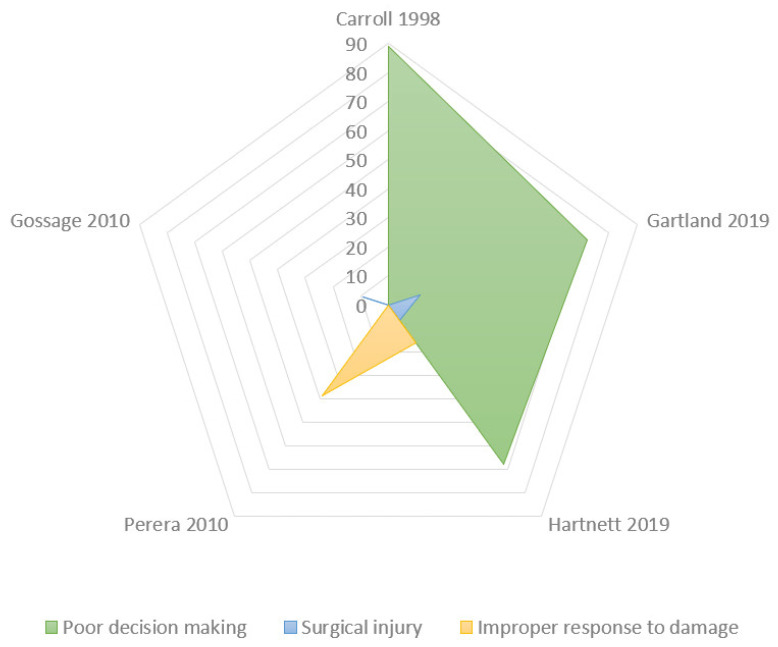
Radar chart: Causes of improper intraoperative surgical performance. The Radar chart shows the biggest polygon (green area) which is the most common indicator of intraoperative surgical performance: poor decision making or misinterpretation (“problematic visualization”). The areas of the smaller polygons (yellow: improper response to damage; blue: surgical injuries) are particularly small to demonstrate the few cases reported.

**Figure 7 jcm-10-05238-f007:**
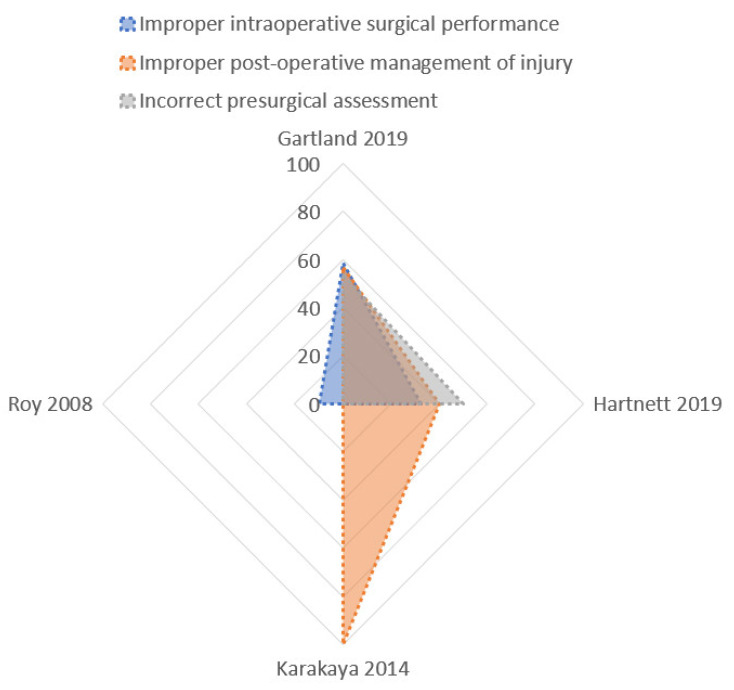
Radar chart: Defensive verdicts in litigations. The largest polygon (pink area) is the most common cause of defensive verdicts in litigations, that is, the improper post-operative management of the injury. The other two areas demonstrate incorrect presurgical assessment (grey area) and improper intraoperative surgical performance (blue area).

**Figure 8 jcm-10-05238-f008:**
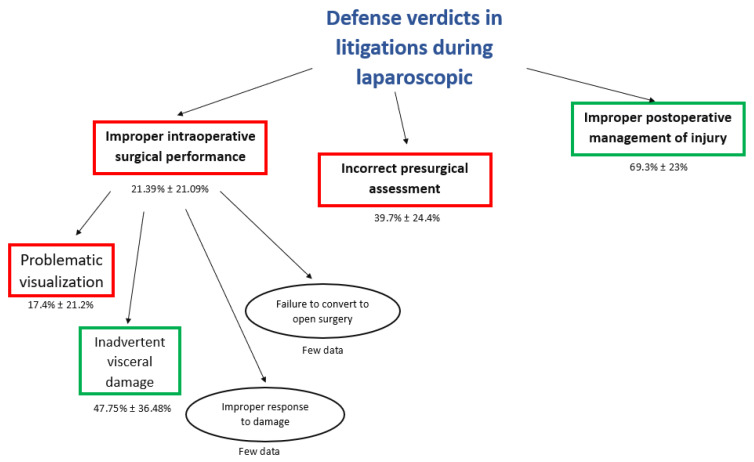
Flowchart of defense verdicts in litigations concerning laparoscopic cholecystectomy.

**Figure 9 jcm-10-05238-f009:**
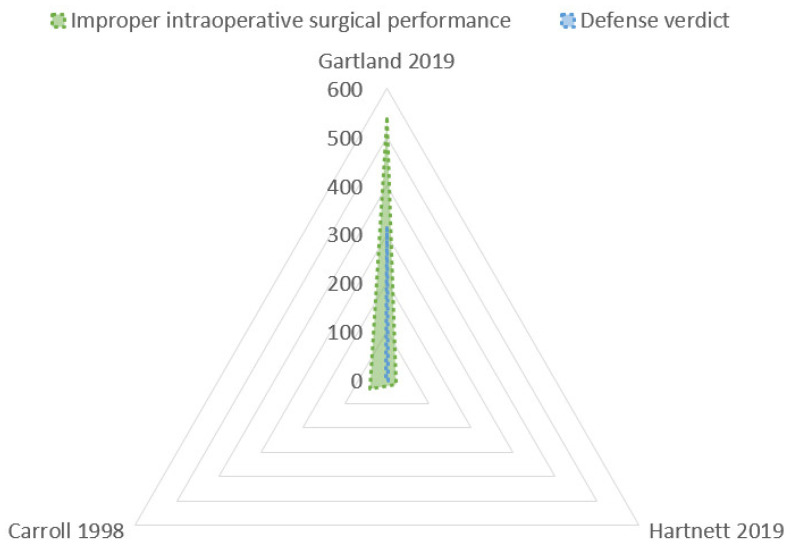
Radar chart: Defensive verdicts in litigations for problematic visualization. The largest polygon (green area: improper intraoperative surgical performances consequent to poor decision making or misinterpretation—“problematic visualization”) is the most common cause of a lower rate of defensive verdicts in litigations (blue area).

**Figure 10 jcm-10-05238-f010:**
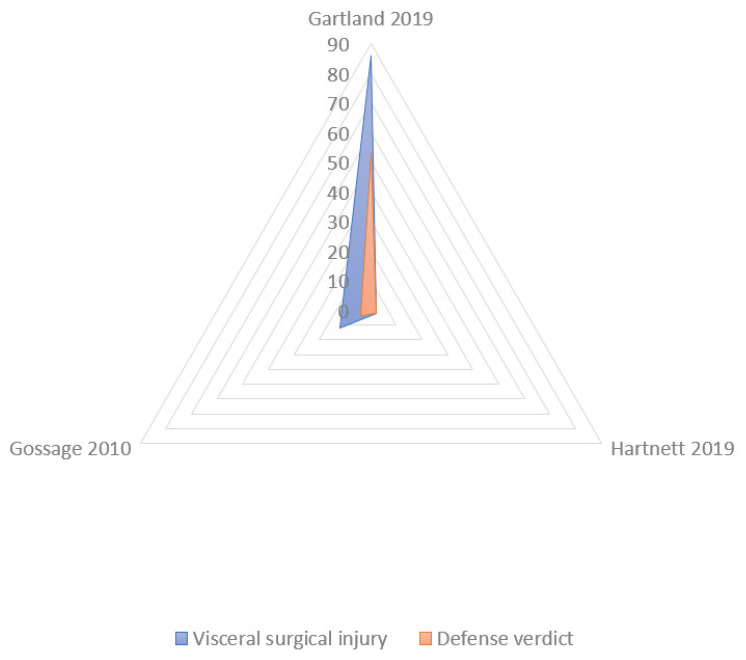
Radar chart: defensive verdicts in litigations for visceral damage. The largest polygon (blue area: visceral surgical injuries) is associated with a high rate of defensive verdicts in litigations (pink area).

**Figure 11 jcm-10-05238-f011:**
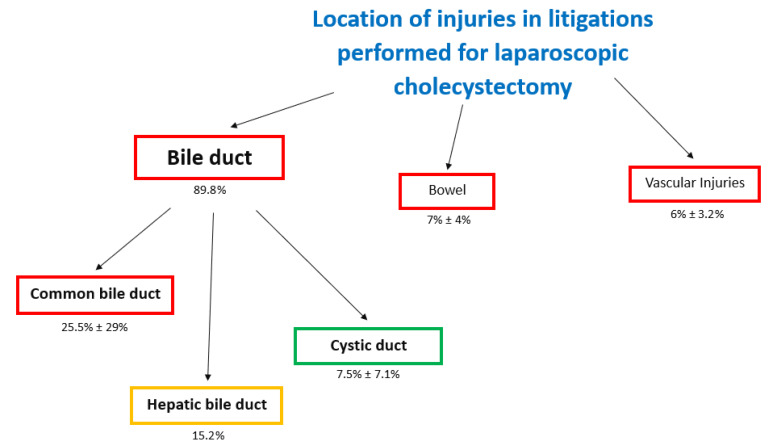
Flowchart of causes of location of injuries in litigations performed for laparoscopic cholecystectomy.

**Figure 12 jcm-10-05238-f012:**
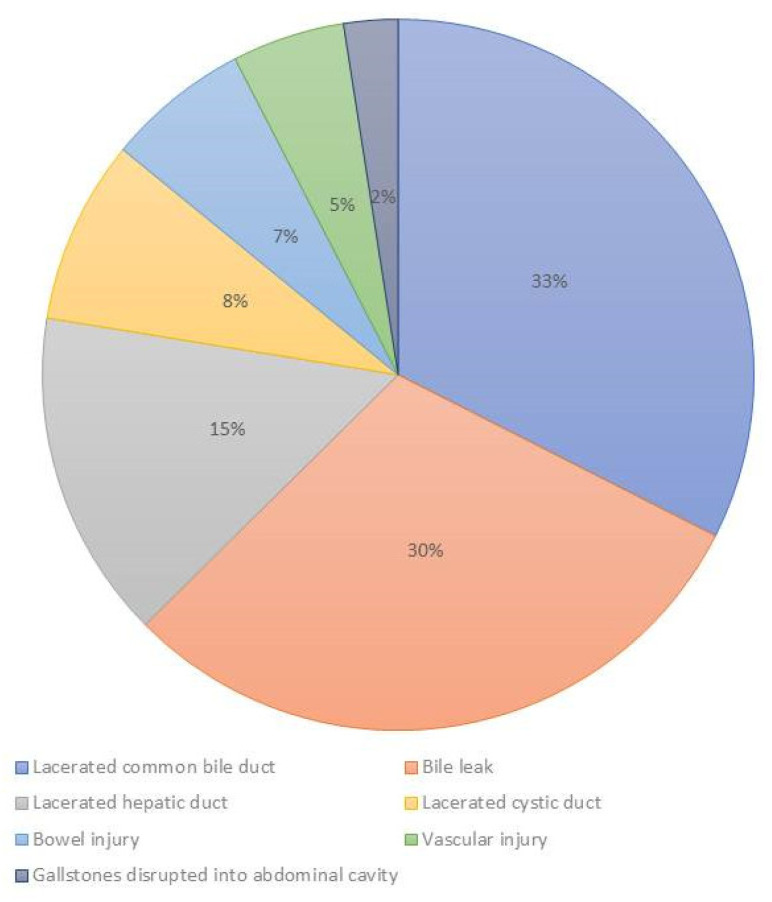
Injuries distribution consequent to laparoscopic cholecystectomy.

**Figure 13 jcm-10-05238-f013:**
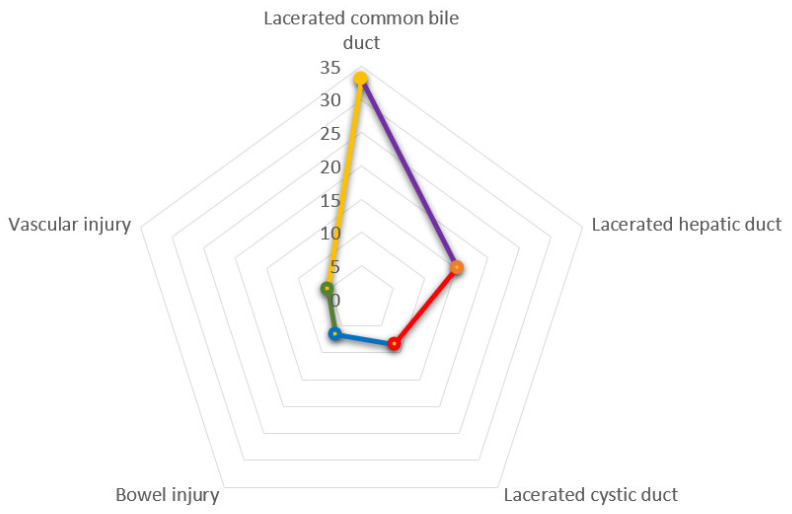
Radar chart of injuries during laparoscopic cholecystectomy. The yellow line shows that the most frequent injury is the laceration of the common bile duct; the other most frequent lesion is the injuries of hepatic duct.

**Figure 14 jcm-10-05238-f014:**
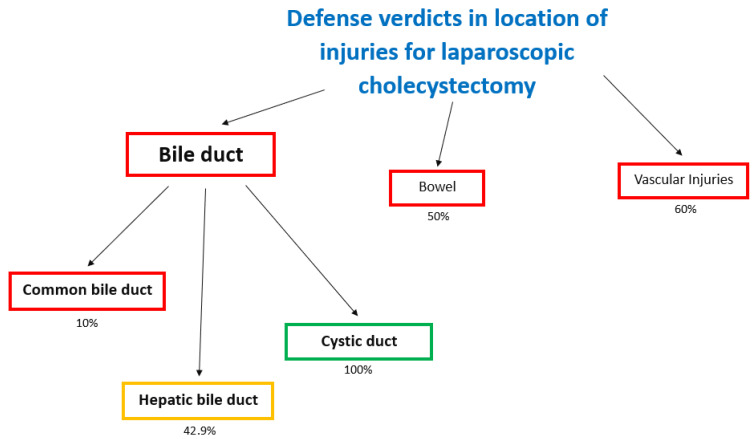
Flowchart of causes of defense verdicts in location of injuries for laparoscopic cholecystectomy.

**Table 1 jcm-10-05238-t001:** Included studies.

Author	Nations	Time of Recruitment	Database	Number of Claims for Laparoscopic Cholecystectomy Reported
Farooq 2020 [24]	U.S.A.	2000–2018	The Westlaw database	231
Hartnett 2019 [25]	U.S.A.	2004–2017	Verdict search database (ALM Media Properties, LLC, New York, NY)	46
Gartland 2019 [26]	U.S.A.	1995–2015	CRICO Strategies’ Comparative Benchmarking System (CBS) database	745
Karakaya 2014 [27]	Turkey	2008–2012	Istanbul Forensic Medicine Institute with request of expert opinion due to bile duct injuries	21
Perera 2010 [28]	U.K.	1992–2009	National Health Service Litigation Authority (NHSLA) for England	67
Alkhaffaf 2010 [29]	U.K.	1995–2009	National Health Service Litigation Authority (NHSLA) for England	418
Scurr 2010 [30]	U.K.	1990–2007	Personal medicolegal experience	151
Gossage 2010 [5]	U.K.	1995–2008	National Health Service Litigation Authority (NHSLA) for England	300
Roy 2009 [17]	U.K.	2000–2005	National Health Service Litigation Authority (NHSLA) for England	64
De Reuver 2008 [31]	Netherland	1994–2006	Dutch Arbitration System for Medical Malpractice	133
McLean 2006 [32]	U.K.	1999–2004	Lexis-Nexis database “Jury Verdict and Settlements, Combined”	104
Carroll 1998 [33]	U.S.A.	1990–1996	NR	46
Chandler 1997 [34]	U.S.A.	1989–1993	Survey of 31 companies of the Physician Insurers Association of America	36
Kern 1997 [35]	U.S.A.	1993–1996	Lexis-Nexis database “Jury Verdict and Settlements, Combined”	44

**Table 2 jcm-10-05238-t002:** Phases of care in which the request of claim for malpractice occurred.

Phase of Care	Incidence	Plaintiff Verdicts (% of Phase)	Arbitration (% of Phase)	Settlement (% of Phase)	Mixed (% of Phase)	Defense Verdicts (% of Phase)	Author
Improper intraoperative surgical performance	48.9% (113/249)	NR	NR	NR	NR	NR	Farooq 2020 [24]
	67.4% (31/46 cases)	51.6%	3.2%	6.5%	6.5%	32.3%	Hartnett 2019 [25]
	72.21% (538/745 cases)	NR	NR	NR	NR	58.4%	Gartland 2019 [26]
	80% (64/83 cases)	NR	NR	NR	NR	10%	Roy 2009 [17]
	80% (37/46 cases)	NR	NR	NR	NR	NR	Carroll 1998 [33]
Improper post-operative management of injury	12.6% (29/249cases)	NR	NR	NR	NR	NR	Farooq 2020 [24]
	10.9% (5/46 cases)	40%	20%	0	0	40%	Hartnett 2019 [25]
	11.14% (83/745 cases)	NR	NR	NR	NR	56.81%	Gartland 2019 [26]
	68,65% (46/67)	NR	NR	NR	NR	NR	Perera 2010 [28]
	95.3% (20/21)	NR	NR	NR	NR	100%	Karakaya 2014 [27]
Incorrect presurgical assessment	2.6% (6/249 cases)	NR	NR	NR	NR	NR	Farooq 2020 [24]
	8.7% (4/46 cases)	25%	0	25%	0	50%	Hartnett 2019 [25]
	5.9% (44/745 cases)	NR	NR	NR	NR	63.6%	Gartland 2019 [26]
Failure to treat injury	3 cases (6.5%)	0	0	0	0	3 cases (100%)	Hartnett 2019 [25]
	8 cases (8.9%)	NR	NR	NR	NR	8 cases (100%)	Carroll 1998 [33]
Unnecessary surgery	1.6% (4/249 cases)	NR	NR	NR	NR	NR	Farooq 2020 [24]
	6.5% (3/46 cases)	33.3%	0	0	0	66.7%	Hartnett 2019 [25]
	1.3% (10/745 cases)	NR	NR	NR	NR	NR	Gartland 2019 [26]

**Table 3 jcm-10-05238-t003:** Improper intraoperative surgical performance in which malpractice occurred.

	Type of Error	Incidence	Plaintiff Verdicts	Arbitration	Settlement	Mixed	Defense Verdicts	Database	Author
Poor decision making or misinterpretation	“Not performing a cholangiogram” “Surgeons misconception about biliary anatomy” “Poor dissection—no critical view obtained” “Surgery related—improper performance”	67.7%, (21/31 cases)	66.7%	4.7%	9.5%	0	19%	Verdict search database (ALM Media Properties, LLC, New York, NY)	Hartnett 2019 [25]
		89.1%, (41/46 cases)	NR	NR	NR	NR	2,4%	NR	Carroll 1998 [33]
		72.21% (538/745 cases)	NR	NR	NR	NR	58,4%	CRICO Strategies’ Comparative Benchmarking System (CBS) database	Gartland 2019 [26]
Surgical injury	Visceral inadvertent damage	6.5%, (2/31 cases)	0	0	0	0	100%	Verdictsearch database (ALM Media Properties, LLC, New York, NY)	Hartnett 2019 [25]
		11.54% (86/745 cases)	NR	NR	NR	NR	60.2%	CRICO Strategies’ Comparative Benchmarking System (CBS) database	Gartland 2019 [26]
		9%	NR	NR	NR	NR	29%	NHS Litigation Authority (NHSLA) database	Gossage 2010 [5]
Failure to convert to open surgery	Failure to convert versus immediate conversion to the open approach	9.7% (3/31)	0	0	0	0	100%	Verdictsearch database (ALM Media Properties, LLC, New York, NY)	Hartnett 2019 [25]
		37,3% (25/67)	NR	NR	NR	NR	NR	NHS litigation authority (NHSLA)	Perera 2010 [28]
Improper response to damage	“Ill-judged attempts to control bleeding”	16.1% (5/31)	40%, (2/5 cases)	0	0	40%, (2/5 cases)	20%, (1/5 cases)	Verdictsearch database (ALM Media Properties, LLC, New York, NY)	Hartnett 2019 [25]
	Immediate repair performed by referring surgeon vs HPB outreach team	38.46% (5/13)	NR	NR	NR	NR	NR	NHS litigation authority (NHSLA)	Perera 2010 [28]

## Data Availability

The searching for relevant studies was performed on PubMed, SCOPUS and on Web of Science from January 1991 to January 2021. The checklist of PRISMA-ScR (Preferred Reporting Items for Systematic reviews and Meta-Analyses extension for Scoping Reviews) was followed as SDC 1. The results are available as request to the corresponding Author.

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
