# Peer review of "Injuries during Laparoscopic Cholecystectomy: A Scoping Review of the Claims and Civil Action Judgements"

_jcm, 2021, doi:10.3390/jcm10225238_

Round 1

Reviewer 1 Report

I have carefully read the manuscriot from the topic Injuries during Laparoscopic Cholecystectomy: A Scoping Review of the Claims and Civil Action Judgments

I think it contains interesting innovative insights about the safety of care that need to be known by health professionals, so I recommend its publication. I think it is possible to improve the text, for example only a brief reference is made to Law no. 24 on the safety of care in Italy. This point is important, the authors should better explain why the study of claims and civil Action Judgments is fundamental in this law and in risk management policies. I suggest this paper for increase references:
Analysis of inadequacies in hospital care through medical liability litigation
Russa, R.L.Viola, R.V.D’errico, S., ...Frati, P.Fineschi, V.
International Journal of Environmental Research and Public Health2021, 18(7), 3425  
Risk management and treatment of coagulation disorders related to COVID-19 infection
Zanza, C.Racca, F.Longhitano, Y., ...Volonnino, G.Russa, R.L. International Journal of Environmental Research and Public Health2021, 18(3), pp. 1–16, 1268

Reviewer 2 Report

The authors performed a review of medico-legal claims related to the laparoscopic cholecystectomy injuries during surgery for gallbladder stones in the international literature over the past three decades. They concluded that during laparoscopic cholecystectomy, the most common cause of claims, associated with lower rate of defense verdict, was the improper intraoperative surgical performance. The decision to take legal action was determined often for poor communication after the original incident.

Overall nice study, I read it with interest. There are several methodological issues that need to be addressed before any favorable decision should be made:

  1. Introduction – Please add your hypothesis in one or two lines.
  2. Introduction – The authors stated that recent studies have reported an increase in BDI up to 0.8% with minimal access surgery which is much higher than the previous reported rates for open cholecystectomy which was between 0.2-0.3% (3). This is very important and good observation. Furthermore due to increased pediatric cholelithiasis and laparoscopies cholecystectomies in pediatric patients the authors should add that this rates in pediatric centers are slightly higher, ranging from 0.5 – 2.5%, probably due to lesser experience of pediatric surgeons with this type of surgeries (REFERENCE: Gallbladder disease in children: A 20-year single-center experience. Indian Pediatr. 2019;56(5):384-386.)
  3. Materials and methodsDefinition of scoping review is not required. Please remove lines from 107-113. Rearrange the reference accordingly.
  4. Materials and methods –Although scoping reviews are different from systematic reviews, it is better to have an optimal database search. For an efficient database search, it is essential to search in atleast four databases. Please see the reference: (Bramer, W.M.; Rethlefsen, M.L.; Kleijnen, J.; Franco, O.H. Optimal database combinations for literature searches in systematicreviews: A prospective exploratory study.Syst. Rev.2017,6, 245.). What was the reason to include only three databases in the search strategy?
  5. Whether it is systematic review or scoping review, it is BETTER to provide a methodological quality and risk of bias assessment. Please use any scoring scale: Downs and Black/Newcastle-Ottawa/ROBINS-I to improve the overall quality of the manuscript.
  6. Results – Table 2 is redundant. Please include the reasons in Figure 1 only.
  7. Results – Again Figure 2 is not required. The country of origin has already been depicted in the Table 1.
  8. –The radar charts are well presented. However, it will be cumbersome for the readers. Please simplify the figures or include one-line legend in each radar chart.
  9. Discussion is very long. It has too much irrelevant info which is outside the interest of this article. Please remove the unnecessary sections.
  10. Limitations of the study need to be mentioned in detail. The authors have mentioned a paragraph in line no 244-249 (page no 12). However, the overall limitations of this manuscript need to be mentioned.
  11. Conclusion is also very lengthy, it should be simplified in one shorter paragraph.

Round 2

Reviewer 2 Report

The authors revised the manuscript according to suggestions and comments of the reviewers. The manuscript has been significantly improved and can be accepted after one minor revision.

Minor objection - The authors added fourth database (literature search), and as I can see in track changes they delete it. Four databases is mandatory. Please revise.
